# Novel Two-Slope Equations to Predict Amino Acid Concentrations Using Crude Protein Concentration in Soybean Meal

**Su A Lee** [1,2]**, Chan Sol Park** [3] **and Beob Gyun Kim** [1,]*

1    Department of Animal Science and Technology, Konkuk University, Seoul 05029, Korea; sualee2@illinois.edu
2    Department of Animal Sciences, University of Illinois, Urbana, IL 61801, USA
3    Department of Animal Sciences, Purdue University, West Lafayette, IN 47907, USA; park765@purdue.edu
*    Correspondence: bgkim@konkuk.ac.kr; Tel.: +82-2-2049-6255

**Abstract:** Amino acid (AA)-to-crude protein (CP) ratios in soybean meal (SBM) may be different for different sources of SBM depending on the presence of additional hulls. Therefore, this study was conducted to develop novel two-slope equations to predict the concentrations of AAs in SBM using CP as an independent variable. Regression analyses were performed with each AA in SBM as the dependent variable and the CP as the independent variable. Among all AAs, the predicted Lys in SBM (% dry matter (DM)) was: Lys = $3.19 - 0.026 \times (51.88 - CP)$ where CP < 51.88% DM and Lys = $3.19 + 0.072 \times (CP - 51.88)$ where CP > 51.88% DM with $R^2 = 0.51$ and $p < 0.001$. In conclusion, the novel equations provided reasonable estimates of the AA concentrations from different ranges of CP in SBM.

**Keywords:** amino acids; broken-line analysis; correlation; crude protein; prediction equation; regression; soybean meal

## 1. Introduction

Soybean meal (SBM) is a byproduct of extracting oil from soybeans and this defatted meal is one of the amino acid (AA) sources for livestock and poultry. The SBM, defined as "soybean seeds without hulls meal solvent extracted" [1] (IFN 5-04-612), is widely used and the average crude protein (CP) concentration is 53.05% on a dry matter (DM) basis [2]. On the other hand, soybean hulls with an average of 10.27% CP on a DM basis [1] (IFN 1-04-560) are often added to SBM at the end of SBM production, resulting in the dilution of AAs in SBM due to the relatively low AA contents in soybean hulls.

The information on the concentrations of AAs in SBM is important for accurate formulation of animal feeds. However, determination of the AA concentration in SBM is generally more laborious and costly than that of CP. Therefore, a simple linear regression analysis with one slope has been used to predict the AA concentrations in SBM [3,4]. Concentrations of AAs and CP in SBM have positive linear correlations [5], but the magnitude of AA concentration changes resulting from CP deviation varies depending on the presence of soybean hulls in SBM, due to the different AA-to-CP ratios between soybean hulls and SBM. In other words, the AA-to-CP ratio in SBM is affected by the addition of soybean hulls to SBM after dehulling and solvent extraction processes. However, the influence of soybean hull inclusion in SBM products on the AA concentrations is not reflected in the simple linear equations [3,4]. To bridge this gap, we developed novel two-slope equations to predict the AA concentrations from CP in SBM.

## 2. Materials and Methods

### 2.1. Soybean Meal Samples

A dataset comprising 192 SBM samples was used. The SBM samples were from Argentina (*n* = 3), Brazil (*n* = 47), China (*n* = 60), India (*n* = 70), and the United States

(*n* = 12). Among the 192 SBM sources, the AA composition data were available only for 64 SBM samples. The average, standard deviation (SD), and coefficient of variation (CV) were calculated for all analyzed nutrients.

*2.2. Chemical Analysis*

The SBM were analyzed for moisture (method 930.15), CP (method 990.03), ether extract (EE; method 2003.05), and ash (method 942.05) as described by the AOAC [6]. Crude fiber (CF) was analyzed using the Ankom filter bag technique (Ankom Technology, Macedon, NY, USA) as described by the AOCS [7] (method Ba 6a-05). Amino acid concentrations were analyzed as described by the AOAC [6]. Soybean meal samples were hydrolyzed with 6 *N* HCl for 24 h at 110 °C (method 994.12) for the analysis of AA except for sulfur-containing AA. For Met and Cys, SBM samples were analyzed as methionine sulfone and cysteic acid after cold performic acid oxidation before the acid hydrolysis.

*2.3. Statistical Analysis*

To predict concentrations of AA in different sources of SBM, a linear analysis to generate the simple linear equations (PROC REG; SAS, SAS Inst. Inc., Cary, NC, USA) and a linear broken-line analysis [8] (PROC NLIN; SAS, SAS Inst. Inc., Cary, NC, USA) to generate the novel two-slope prediction equations [9] were performed with the AA concentrations in the SBM as dependent variables, and the CP concentrations as the independent variable. Models were validated based on the root mean square error (RMSE), coefficient of determination ($R^2$), standard error, and *p*-value. An alpha level of statistical significance was set at 0.05 and tendency was considered at $0.05 \leq p < 0.10$.

## 3. Results

*3.1. Nutrient Composition in Different Sources of Soybean Meal*

The concentration of moisture in the 192 sources of SBM ranged from 9.38 to 13.41% DM (average = 12.05; CV = 5.3%; Table 1). The CP concentration in SBM had less variation compared with CF, EE, and ash (average = 51.88% DM; CV = 2.1%). The average CF concentration in SBM was 6.45% DM. The EE concentration in SBM was 1.50% DM and had the greatest CV value (CV = 44.4%) compared to other nutrients. The average ash concentration was 7.35% DM (CV = 13.5%). Analyzed Lys, Met, and Thr in the 64 SBM sources were 3.22, 0.68, and 2.02% DM, respectively. The CV for AA concentrations in SBM ranged from 2.0 to 6.0%.

**Table 1.** Chemical composition of soybean meal, dry matter basis [1].

| Item, % | Average | SD [2] | Minimum | Maximum | CV [2], % |
|---------|---------|--------|---------|---------|-----------|
| Moisture | 12.05 | 0.63 | 9.38 | 13.41 | 5.3 |
| Crude protein | 51.88 | 1.07 | 49.04 | 54.61 | 2.1 |
| Crude fiber | 6.45 | 1.13 | 3.65 | 9.09 | 17.6 |
| Ether extract | 1.50 | 0.67 | 0.35 | 5.62 | 44.4 |
| Ash | 7.35 | 0.99 | 5.56 | 11.44 | 13.5 |
| Indispensable amino acids | | | | | |
| Arg | 3.71 | 0.07 | 3.52 | 3.86 | 2.0 |
| His | 1.35 | 0.03 | 1.29 | 1.43 | 2.2 |
| Ile | 2.37 | 0.09 | 2.10 | 2.53 | 3.7 |
| Leu | 4.00 | 0.09 | 3.81 | 4.30 | 2.1 |
| Lys | 3.22 | 0.07 | 3.09 | 3.35 | 2.1 |
| Met | 0.68 | 0.03 | 0.63 | 0.76 | 5.0 |
| Phe | 2.65 | 0.16 | 2.51 | 3.81 | 6.0 |
| Thr | 2.02 | 0.06 | 1.92 | 2.15 | 2.9 |
| Val | 2.47 | 0.08 | 2.21 | 2.57 | 3.3 |

**Table 1.** *Cont.*

| Item, % | Average | SD [2] | Minimum | Maximum | CV [2], % |
|---|---|---|---|---|---|
| Dispensable amino acids | | | | | |
| Ala | 2.25 | 0.05 | 2.13 | 2.35 | 2.0 |
| Asp | 5.84 | 0.14 | 5.60 | 6.20 | 2.3 |
| Cys | 0.71 | 0.04 | 0.63 | 0.78 | 5.3 |
| Glu | 9.43 | 0.22 | 8.91 | 9.78 | 2.3 |
| Gly | 2.19 | 0.04 | 2.10 | 2.29 | 2.0 |
| Pro | 2.58 | 0.07 | 2.45 | 2.75 | 2.8 |
| Ser | 2.55 | 0.09 | 2.40 | 2.77 | 3.4 |
| Tyr | 1.72 | 0.07 | 1.53 | 1.85 | 4.0 |

[1] Each mean represents 192 observations except amino acid compositions ($n = 64$) and Tyr ($n = 63$). [2] SD = standard deviation; CV = coefficient of variation.

Average values for Lys-to-CP, Met-to-CP, Thr-to-CP, and Val-to-CP ratios in the different sources of SBM were 0.062, 0.013, 0.039, and 0.047, respectively (Table 2). The ratios between dispensable AAs and CP ranged from 0.014 (Cys) to 0.181 (Glu).

**Table 2.** Amino acid (AA)-to-crude protein (CP) ratio in soybean meal [1].

| Item, % | Average | SD [2] | Minimum | Maximum | CV [2], % |
|---|---|---|---|---|---|
| Indispensable AAs-to-CP | | | | | |
| Arg | 0.071 | 0.001 | 0.068 | 0.074 | 1.7 |
| His | 0.026 | 0.001 | 0.025 | 0.027 | 2.0 |
| Ile | 0.046 | 0.001 | 0.042 | 0.048 | 2.9 |
| Leu | 0.077 | 0.002 | 0.074 | 0.080 | 2.0 |
| Lys | 0.062 | 0.001 | 0.060 | 0.064 | 1.6 |
| Met | 0.013 | 0.001 | 0.012 | 0.015 | 4.4 |
| Phe | 0.051 | 0.003 | 0.048 | 0.074 | 6.0 |
| Thr | 0.039 | 0.001 | 0.037 | 0.041 | 3.0 |
| Val | 0.047 | 0.001 | 0.044 | 0.050 | 2.9 |
| Dispensable AAs-to-CP | | | | | |
| Ala | 0.043 | 0.001 | 0.041 | 0.045 | 1.9 |
| Asp | 0.112 | 0.002 | 0.109 | 0.117 | 1.6 |
| Cys | 0.014 | 0.001 | 0.012 | 0.015 | 4.6 |
| Glu | 0.181 | 0.003 | 0.176 | 0.188 | 1.6 |
| Gly | 0.042 | 0.001 | 0.041 | 0.044 | 1.9 |
| Pro | 0.050 | 0.001 | 0.047 | 0.054 | 2.6 |
| Ser | 0.049 | 0.002 | 0.046 | 0.054 | 3.7 |
| Tyr | 0.033 | 0.001 | 0.029 | 0.036 | 4.0 |

[1] Each mean represents 64 observations except Tyr-to-CP ($n = 63$). [2] SD = standard deviation; CV = coefficient of variation.

*3.2. Correlation Coefficients between Crude Protein and Amino Acids and Simple Linear Regressions*

Concentrations of CP in the 64 different sources of SBM were positively correlated ($p < 0.001$) with most AAs except Phe, Thr, Ser, and Tyr (Table 3). The concentrations of Thr and Tyr tended to have positive correlations with CP ($p < 0.10$).

Regression of AA concentrations in SBM against concentrations of CP indicated that the slopes of the regressions were in positive values (Table 4). The coefficient of determination ($R^2$) for the equations ranged from 0.02 (Phe) to 0.48 (Lys).

**Table 3.** Correlation coefficients (r) between crude protein and amino acid concentrations (% on a dry matter basis) [1].

| Item | Crude Protein (r) | *p*-Value |
|---|---|---|
| Indispensable amino acids | | |
| Arg | 0.627 | <0.001 |
| His | 0.517 | <0.001 |
| Ile | 0.622 | <0.001 |
| Leu | 0.506 | <0.001 |
| Lys | 0.693 | <0.001 |
| Met | 0.479 | <0.001 |
| Phe | 0.158 | 0.212 |
| Thr | 0.224 | 0.075 |
| Val | 0.496 | <0.001 |
| Dispensable amino acids | | |
| Ala | 0.498 | <0.001 |
| Asp | 0.741 | <0.001 |
| Cys | 0.522 | <0.001 |
| Glu | 0.726 | <0.001 |
| Gly | 0.523 | <0.001 |
| Pro | 0.437 | <0.001 |
| Ser | 0.068 | 0.593 |
| Tyr | 0.213 | 0.093 |

[1] Each mean represents 64 observations except Tyr (*n* = 63).

**Table 4.** Simple linear regression equations to predict indispensable amino acids (% on a dry matter basis) using concentration of crude protein (% on a dry matter basis) in soybean meals (*n* = 64).

| Item | Regression Coefficient Parameter | | Statistical Parameter | | |
|---|---|---|---|---|---|
| | Intercept | Slope (Crude Protein) | RMSE [1] | R-Square | *p*-Value |
| Arg | 1.05 | 0.051 | 0.058 | 0.41 | <0.001 |
| SE [2] | 0.41 | 0.008 | | | |
| *p*-value | 0.013 | <0.001 | | | |
| His | 0.468 | 0.017 | 0.026 | 0.27 | <0.001 |
| SE | 0.19 | 0.004 | | | |
| *p*-value | 0.014 | <0.001 | | | |
| Ile | −0.695 | 0.059 | 0.069 | 0.39 | <0.001 |
| SE | 0.49 | 0.009 | | | |
| *p*-value | 0.161 | <0.001 | | | |
| Leu | 1.54 | 0.047 | 0.075 | 0.26 | <0.001 |
| SE | 0.53 | 0.01 | | | |
| *p*-value | 0.005 | <0.001 | | | |
| Lys | 0.527 | 0.052 | 0.050 | 0.48 | <0.001 |
| SE | 0.36 | 0.007 | | | |
| *p*-value | 0.144 | <0.001 | | | |
| Met | −0.281 | 0.019 | 0.030 | 0.25 | <0.001 |
| SE | 0.21 | 0.004 | | | |
| *p*-value | 0.191 | <0.001 | | | |
| Phe | 1.29 | 0.026 | 0.160 | 0.02 | 0.233 |
| SE | 1.14 | 0.022 | | | |
| *p*-value | 0.262 | 0.233 | | | |
| Thr | 1.34 | 0.013 | 0.057 | 0.05 | 0.095 |
| SE | 0.40 | 0.008 | | | |
| *p*-value | 0.002 | 0.095 | | | |
| Val | 0.11 | 0.045 | 0.070 | 0.27 | <0.001 |
| SE | 0.50 | 0.01 | | | |
| *p*-value | 0.825 | <0.001 | | | |

[1] RMSE = root mean square error. [2] SE = standard error.

### 3.3. Novel Equations to Predict Amino Acids Using Crude Protein in Soybean Meal

The concentration of CP in the 64 sources of SBM was used as an independent variable to predict the indispensable AAs using broken-line models, but the convergence criteria were met only for the concentrations of His, Ile, Lys, Met, and Val (Figure 1) among the nine indispensable AA. The concentrations of five indispensable AA were predicted by different CP ranges and the average of the five breakpoints was 51.60% CP on a DM basis.

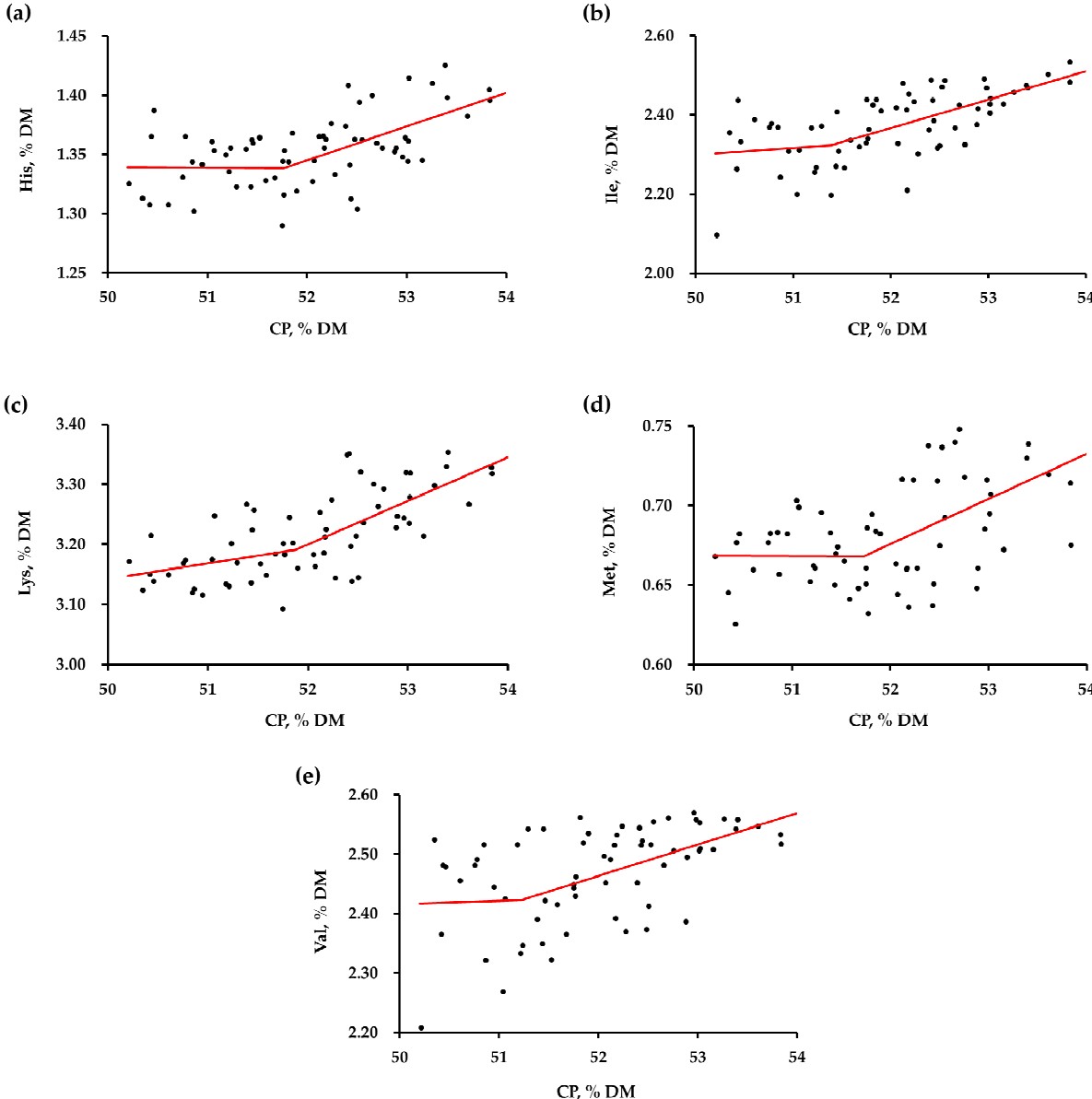

**Figure 1.** Prediction equations for His, Ile, Lys, Met, and Val (% dry mater (DM)) using crude protein (% DM) in soybean meal (*n* = 64). Linear broken-line models represented by solid lines indicate: (**a**) His = 1.34 + 0.0006 × (51.76 − CP) where CP < 51.76 and His = 1.34 + 0.028 × (CP − 51.76) where CP > 51.76 ($R^2$ = 0.32, root mean square error (RMSE) = 0.025, and *p* < 0.001); (**b**) Ile = 2.32 − 0.018 × (51.39 − CP) where CP < 51.39 and Ile = 2.32 + 0.072 × (CP − 51.39) where CP > 51.39 ($R^2$ = 0.40, RMSE = 0.069, and *p* < 0.001); (**c**) Lys = 3.19 − 0.026 × (51.88 − CP) where CP < 51.88 and Lys = 3.19 + 0.072 × (CP − 51.88) where CP > 51.88 ($R^2$ = 0.51, RMSE = 0.049, and *p* < 0.001); (**d**) Met = 0.67 − 0.0001 × (51.73 − CP) where CP < 51.73 and Met = 0.67 + 0.028 × (CP − 51.73) where CP > 51.73 ($R^2$ = 0.27, RMSE = 0.030, and *p* < 0.001); (**e**) Val = 2.42 − 0.005 × (51.23 − CP) where CP < 51.23 and Val = 2.42 + 0.053 × (CP − 51.23) where CP > 51.23 ($R^2$ = 0.26, RMSE = 0.072, and *p* < 0.001).

## 4. Discussion

Even though hulls are removed from the soybeans before extraction of the oil, soybean hulls are sometimes added back to the dehulled SBM after oil extraction [10,11]. As soybean hulls contain lower concentrations of AAs compared with dehulled SBM, the addition of soy hulls to dehulled SBM results in the dilution of AAs. As CP in SBM increase, concentrations of AAs linearly increase [5]. Additionally, the AA-to-CP ratio in SBM also changes when soybean hulls are added back as AA-to-CP ratios are not constant between SBM and soybean hulls. Therefore, the consideration of the deviation of AA-to-CP ratios between SBM and soybean hulls is important in the equations for predicting AA in SBM products.

The concentrations of moisture, CP, EE, and ash in the 192 sources of SBM were within the range of the values in the literature [2,3,12–14]. The concentrations of EE in the 192 samples of SBM had the greatest CV value compared with the other chemical compositions. The reason for this observation may be due to the variation in solvent extraction qualities among production plants. In addition, the CV of CF concentrations among SBM sources was the second greatest among chemical compositions. This may reflect the wide range of SBM sources used in the current study, which encompassed both dehulled SBM and SBM with hulls. Values for the CV of indispensable AAs were close to the CV among 31 samples of SBM reported by Cromwell et al. [15]. Despite the wide range of SBM sources, the ratio of AAs and CP was relatively consistent with low CV values, resulting in strong correlations between most AAs and CP concentrations.

The amino group ($NH_2$) of AAs reacts with reducing sugar in the presence of heat and moisture to produce Amadori compounds and melanoidins [16]. This reaction is called the Maillard reaction, which leads to changes in color and sensory properties of feed ingredients. The Amadori compounds and melanoidins are biologically not available and the advanced Maillard reaction products can also react with AAs and make them unavailable in the animals' body [17,18]. Among all AAs, Lys is usually the AA that is most susceptible to heat damage because Lys has an amino group in the side chain (i.e., the epsilon amino group). Therefore, in most cases, the Lys-to-CP ratio indicates the degree of heat damage in SBM because, with a constant CP concentration, the concentration of Lys is reduced as SBM is heat damaged. Previous data demonstrated that the Lys:CP was reduced to less than 6.00 in heat-damaged SBM [17]. The observation that the Lys:CP in the SBM sources used in this experiment ranged from 6.0 to 6.4 indicated that very little heat damage took place during the process of producing those SBM sources.

Linear regression has been widely used for the prediction of the time- and cost-consuming criteria, including in vivo data [5,19–24] or data from a relatively tedious analysis for the feed ingredients [3,4,25,26]. More time and expenses may be required to determine the AA than the CP concentrations in feed ingredients. Additionally, the CP concentrations, as well as the AA concentrations in the same type of SBM, vary among the sources [27] because the characteristics of plants are reflected mostly by genetic [28] and various environmental factors [29]. The CP in the SBM sources suggested in AMINODat [3] ranged from 45.23% to 59.55% DM and the $R^2$ values for the predictions of AAs were 0.62 (Arg), 0.46 (His), 0.58 (Ile), 0.74 (Leu), 0.42 (Lys), 0.61 (Phe), 0.49 (Thr), 0.45 (Trp), and 0.52 (Val), most of which were greater than the $R^2$ values of the simple linear regressions in Table 4. According to the reports by USSEC [4], the CP concentrations in 403 sources of SBM ($48.4 < CP < 58.0\%$ DM) were able to predict the Lys concentrations with fairly reasonable estimates ($R^2 = 0.72$). The greater $R^2$ values in the previous studies may be due to the greater number of observations and the wider range of independent variables compared with the present study. Cromwell et al. [15] also reported prediction equations with greater $R^2$ values, ranging from 0.54 to 0.94, to estimate the concentration of indispensable AAs in 31 SBM samples, including both dehulled SBM and SBM with hulls. The discrepancy between the current study and that of Cromwell et al. [15] may be due to the difference in origin of SBM sources. The 61 SBM samples used in the current study were collected from around the world, with China and India as two major countries. However, all sources of

SBM used in Cromwell et al. [15] originated in the United States, although SBM samples were collected from various stations in different states.

The prediction equations from the previous studies used single slopes. To overcome the limitations of one-slope equations, novel prediction equations were developed using two-slope broken-line analyses in the present work to reflect the influence of the soybean hull inclusion in SBM on the AA-to-CP ratio. Coefficients of determination ($R^2$) represent the accuracy of prediction, and the values from the novel equations were slightly greater than those from the simple linear regression equations. The average of breakpoints from five equations was close to the mean value for CP concentrations in dehulled SBM and SBM with hulls reported in the NRC (average of 51.3% CP on a DM basis) [2] and Park et al. (average of 50.7% on a DM basis) [30]. This indicates that the prediction equations developed by broken-line analysis appropriately reflect the differences in the AA-to-CP ratio between dehulled SBM and SBM with hulls. In addition, previous studies reported that the standardized ileal digestibility of most AAs in dehulled SBM was not different from that in SBM with hulls [30,31]. Therefore, the changes in the AA-to-CP ratio in SBM due to the presence of soybean hulls observed in the current study may be also useful in estimating the standardized ileal digestible AA concentrations in SBM with wide rages of CP.

## 5. Conclusions

The concentrations of His, Ile, Lys, Met, and Val in SBM in which CP ranges from 49.04% to 54.61% DM can be predicted by the two-slope prediction equations using CP as an independent variable. The average CP of the five breakpoints was 51.60% DM.

**Author Contributions:** Conceptualization, C.S.P. and B.G.K.; validation, C.S.P. and B.G.K.; formal analysis, S.A.L. and C.S.P.; writing—original draft preparation, S.A.L.; writing—review and editing, C.S.P. and B.G.K.; visualization, S.A.L.; supervision, B.G.K.; funding acquisition, B.G.K. All authors have read and agreed to the published version of the manuscript.

**Funding:** The present research was supported by Konkuk University in 2018.

**Institutional Review Board Statement:** Not applicable.

**Data Availability Statement:** The data presented in this study are available on request.

**Conflicts of Interest:** The authors declare no conflict of interest.

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
