# Peer review of "Novel Two-Slope Equations to Predict Amino Acid Concentrations Using Crude Protein Concentration in Soybean Meal"

_agriculture, doi:10.3390/agriculture11040280_

Round 1

Reviewer 1 Report

This manuscript predicts the concentrations of His, Ile, Lys, Met, and Val in SBM by the two-slope prediction equations using CP as an independent variable. The scientific subject, applied methods and the way of describing and interpreting results is suitable to the journal. It is mentioned in the manuscript that only 64 SBM samples have AA composition data, can the other AA concentrations not be detected or what is the reason?

Author Response

Reviewer #1:

This manuscript predicts the concentrations of His, Ile, Lys, Met, and Val in SBM by the two-slope prediction equations using CP as an independent variable. The scientific subject, applied methods and the way of describing and interpreting results is suitable to the journal. It is mentioned in the manuscript that only 64 SBM samples have AA composition data, can the other AA concentrations not be detected or what is the reason?

RESPONSE: Thank you for the encouraging comments on the manuscript. Due to the limited budget, only 64 SBM samples were analyzed for AA. However, the CP concentrations of 64 SBM samples were widely spread out and should be sufficient to represent the population.

Additional: These changes have been indicated in red color font in the revised manuscript.

L 55: In the introduction, changed “… of the animal feeds” to “… of animal feeds.”

L 56-57: Therefore, a simple linear regression analysis has been … in SBM with one slope [3,4]. >> Therefore, a simple linear regression analysis with one slope has been … in SBM [3,4].

L 59: AA concentration change >> AA concentration changes

L 150: In the discussion section, inserted “concentrations” to make “.. CP concentrations … vary …”

L 168: equations … studies have used … >> equations … studies used …

Reviewer 2 Report

The Authors of the manuscript acknowledge that information about the concentration of AA in SBM is important for accurate balance of diets for animals. Additionally, the Authors indicate that the concentration of AA will be affected by the addition of hulls to the SMB. From the data in Table 1, it appears that the SBM analysed had a very different CF content (CV, 17.6%) and EE (CV 44.4%), which indicates the addition of hulls. However, it is not certain what number of hulls and whether the same is always added to SBM. And if so, the equations developed have only a cognitive rather than a practical aspect.

The quality of soybean seeds and thus SBM (AA and CP content) will depend on many variables such as soybean variety, climatic conditions, soil quality and fertilization, agrotechnical treatments, fat residue after solvent extraction, and therefore it is difficult to take them all into account when constructing equations.

The Authors had 192 SBM samples at their disposal, but only 63 samples were used to determine the AA-CP relationship, which resulted in this difference?

Author Response

Reviewer #2:

The Authors of the manuscript acknowledge that information about the concentration of AA in SBM is important for accurate balance of diets for animals. Additionally, the Authors indicate that the concentration of AA will be affected by the addition of hulls to the SMB. From the data in Table 1, it appears that the SBM analysed had a very different CF content (CV, 17.6%) and EE (CV 44.4%), which indicates the addition of hulls. However, it is not certain what number of hulls and whether the same is always added to SBM. And if so, the equations developed have only a cognitive rather than a practical aspect.

RESPONSE: One of the strengths of the present equations is that animal feed producers can estimate amino acid concentrations regardless of the inclusion of soybean hulls in the soybean meal product. Thus, the present models are practically more applicable compared with other simple linear equations in the literature.

The quality of soybean seeds and thus SBM (AA and CP content) will depend on many variables such as soybean variety, climatic conditions, soil quality and fertilization, agrotechnical treatments, fat residue after solvent extraction, and therefore it is difficult to take them all into account when constructing equations.

RESPONSE: I agree with your comments in that it is difficult to consider all factors that affect AA contents in SBM. In addition, animal feed producers do not have all information on SBM, but they mostly have CP data in hand.

The Authors had 192 SBM samples at their disposal, but only 63 samples were used to determine the AA-CP relationship, which resulted in this difference?

RESPONSE: Due to the limited budget, only 64 SBM samples were analyzed for AA. However, the CP concentrations of 64 SBM samples were widely spread out and should be sufficient to represent the population.

Additional changes: These changes have been indicated in red color font in the revised manuscript.

L 55: In the introduction, changed “… of the animal feeds” to “… of animal feeds.”

L 56-57: Therefore, a simple linear regression analysis has been … in SBM with one slope [3,4]. >> Therefore, a simple linear regression analysis with one slope has been … in SBM [3,4].

L 59: AA concentration change >> AA concentration changes

L 150: In the discussion section, inserted “concentrations” to make “.. CP concentrations … vary …”

L 168: equations … studies have used … >> equations … studies used …